# Hybrid Structure of TiO$_2$-Graphitic Carbon as a Support of Pt Nanoparticles for Catalyzing Oxygen Reduction Reaction

**Su-Jin Jang [1,2], Yun Chan Kang [2], Jin-Su Hyun [3], Tae Ho Shin [1,*], Young Wook Lee [4,*] and Kwang Chul Roh [1,*]**

1   Energy and Environmental Division, Korea Institute of Ceramic Engineering and Technology, Jinju 52851, Korea; jsjin@kicet.re.kr
2   Department of Materials Science and Engineering, Korea University, Anam-Dong, Seongbuk-Gu, Seoul 136-713, Korea; yckang@korea.ac.kr
3   Material Business Team, Vina Tech Co., Ltd., Jeonju 54853, Korea; jshyun@vina.co.kr
4   Department of Education Chemistry and Research Institute of National Sciences, Gyeongsang National University, Jinju 52828, Korea
*   Correspondence: ths@kicet.re.kr (T.H.S.); lyw2020@gnu.ac.kr (Y.W.L.); rkc@kicet.re.kr (K.C.R.); Tel.: +82-55-792-2625 (K.C.R.)

**Abstract:** The durability of catalysts in fuel cells is a longstanding issue that needs to be resolved. Catalyst stability of the fuel cell has always been a problem, studies are underway to address them. Herein, to address this issue, we synthesize a hybrid structure consisting of SP carbon (SP) as the graphitic carbon and TiO$_2$ as the metal oxide using a microwave method for use as a support for Pt nanoparticles. Anatase TiO$_2$ and Pt nanoparticles with sizes of ~5 and 3.5 ± 1.4 nm, respectively, are uniformly dispersed on a modified graphitic SP carbon support (Pt-TiO$_2$-SP). This supported Pt catalyst exhibits significantly improves durability in the oxygen reduction reaction (ORR). Furthermore, the Pt-TiO$_2$-SP carbon hybrid catalyst manifests superior electrocatalytic stability and higher onset potential in ORR than those exhibited by Pt-SP carbon without TiO$_2$. Pt-TiO$_2$-SP exhibits an activity loss of less than 68 mV after 5000 electrochemical cycles, whereas an activity loss of ~100 mV is observed for Pt-SP carbon in a stability test. These results suggest that the strong metal–support interaction in TiO$_2$-supported Pt catalyst significantly enhances the activity of Pt nanocatalyst.

**Keywords:** TiO$_2$-carbon hybrid; catalyst support; durability; fuel cell; graphitic carbon

## 1. Introduction

Proton exchange membrane fuel cells (PEMFCs) have attracted considerable attention due to their high power density, low working temperature, and energy conversion efficiency. Although non-metal catalysts have recently been developed for PEMFCs, Pt remains an attractive candidate catalyst due to its considerable catalytic activity in both the anodic and cathodic reactions [1,2]. However, the application of Pt as a catalyst faces two major obstacles, namely degradation and low durability [3–5]. The redeposition of Pt catalyst particles and corrosion of the support result in the degradation of PEMFC performance because the catalytic activity is directly proportional to the available surface area of the Pt catalyst [6,7]. Among the known causes for the low durability of electrocatalysts, the decrease in the electrochemical surface area (ECSA) on account of the corrosion of the catalyst support and agglomeration of the deposit Pt nanoparticles on supports have been acknowledged as the most important problems.

Carbon is the most widely used support for PEMFCs due to its high electrical conductivity, easy accessibility, affordability, and relatively high electrochemical stability [6,8]. The key favorable properties of supported carbons are their pore sizes with some amount of micropores, high degree of graphitization, and presence of oxygen functional groups [8–11]; however, the supported carbon corrodes to form carbon dioxide under the operating conditions, resulting in deteriorated mass transport to the reaction sites that leads to degraded

PEMFC performance. In addition, the degradation of carbon gives rise to the appearance of Pt particles that detach from the support and form aggregates, thereby decreasing the rate of the active sites. Consequently, graphitized carbon materials are proposed to solve the carbon corrosion. Since carbon nanotubes that are a representative structure of graphitization have high chemical inertness and are difficult to disperse in solution, they must be handled using a chemical treatment. Although Vulcan carbon is widely used as a support in fuel cells, it has poor stability during long-term operation. Therefore, the properties of the carbon support are vital because they affect the electrical conductivity and durability of membrane electrode assemblies.

It has been found that the strong metal–support interaction (SMSI) effect can enhance the activity of Pt nanocatalysts. To improve the stability of catalyst support, several investigations have focused on alternative electrocatalyst supports such as the $TiO_2$, $SnO_2$, and $WO_3$ transition metal oxides in order to develop carbon-free catalysts [12–14]. Metal oxide supports provide electrons to the Pt nanoparticles and affect the unfilled d-band states of Pt, facilitating the activity of the supported catalysts. The so-called SMSI between the metal nanocatalyst and transition metal oxide enhances the catalytic activity in Pt-catalyzed reactions [13,15–17]. $TiO_2$ is widely used as a support due to its resistance to corrosion and low cost. However, its low electrical conductivity is a major drawback that limits its performance as a support. Consequently, hybrid materials consisting of $TiO_2$ and carbon have been developed to enhance the electrical conductivity of $TiO_2$ [2,3,12,16–19].

For the synthesis of metal nanoparticles, microwave-assisted synthesis has the advantage of short synthesis time and homogeneous heating of the materials. Although the gel process, solid-state reactions, and hydrothermal method are commonly used for the synthesis of metal oxides particles, these methods have the drawbacks of involving multiple steps, long synthesis time, poor dispersibility, and lack of control particle size. Thus, compared to other methods, microwave-assisted synthesis is favorable because it obtains a narrow particle size distribution, small particle size, and requires only a short time because of the rapid crystal nucleation and growth process.

In this study, Pt-$TiO_2$-carbon was prepared as a high active oxygen reduction reaction (ORR) catalyst due to its SMSI. We employed SP carbon that has a graphitic structure in which the graphene layers adopt a spherical morphology as the carbon support. The hybrid catalyst support was prepared by a simple microwave method from SP carbon composited with $TiCl_4$ to form the anatase phase of $TiO_2$. In this hybrid catalyst, because the SMSI effect can change the binding energy of Pt, the ORR is enhanced due to the enhanced ECSA of the catalyst. We found that the electrochemical properties of the hybrid catalyst synthesized by the simple microwave method led to improved ORR efficiency.

## 2. Results and Discussion

### 2.1. Preparation and Characterization of the $TiO_2$-SP Hybrid

To prevent the corrosion of carbon for PEMFC, the hybrid $TiO_2$-SP structure comprised by graphitic SP carbon and $TiO_2$ was synthesized using a microwave-assisted method. The morphology and size distribution of $TiO_2$ in $TiO_2$-SP were investigated by transmission electron microscopy (TEM). Figure 1a,b shows the TEM images of $TiO_2$-SP, and it is observed that the $TiO_2$ on the SP carbon surface has a rough spherical morphology (~5 nm). The high-resolution (HR)-TEM image of $TiO_2$-SP in Figure 1c shows lattice fringes with a spacing of 0.332 nm, corresponding to the (101) plane of anatase $TiO_2$ [12,15]. The high-angle annular dark-field-scanning TEM (HAADF-STEM) image in Figure 1d shows metal nanoparticles and the corresponding energy-dispersive X-ray spectroscopy (EDS) mapping images in Figure 1e–h shows the distribution of $TiO_2$ in the carbon support. The results reveal that nano-sized $TiO_2$ particles are uniformly distributed on the SP carbon surface, suggesting that the $TiO_2$-SP hybrid was effectively produced by the simple microwave method, and consists of $TiO_2$ nanoparticles uniformly deposited on SP carbon.

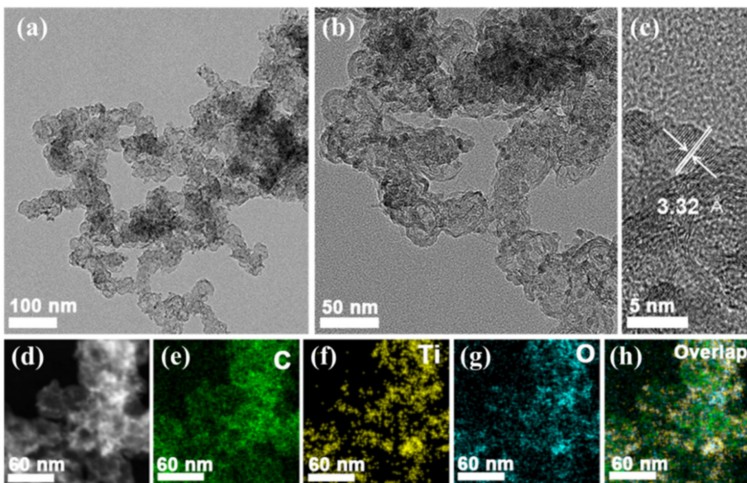

**Figure 1.** (**a,b**) TEM images of TiO$_2$-SP at different magnifications. (**c**) HR-TEM image of TiO$_2$-SP. (**d**) HAADF-STEM image and (**e–h**) the corresponding EDS elemental mapping images of TiO$_2$-SP.

### 2.2. Comparison of the Structures of SP Carbon and Vulcan XC-72

To identify the structure of SP carbon, the Raman spectra of SP carbon (SP) and Vulcan XC-72 (Vulcan) that are used as carbon supports are shown in Figure 2. Vulcan carbon is commonly used as a support in fuel cells; however, its low degree of graphitization is a key issue because it affects the electrical conductivity and stability of the membrane electrode assemblies. The Raman spectra of Vulcan XC-72 and SP carbon in Figure 2a show two prominent bands, namely the D and G bands. The D band located at ~1350 cm$^{-1}$ is related to local defects and disorder, whereas the G band at ~1594 cm$^{-1}$ typically appears due to the in-plane vibration of the sp$^2$-bonded carbon atoms [20,21]. The specific surface area of SP (440 m$^2$ g$^{-1}$) is almost twice higher than that of Vulcan (250 m$^2$ g$^{-1}$) because SP was treated by a high-temperature activation process (Figure S1). The changes in the structure of SP due to the activation process also give rise to the higher intensity of the D band of SP compared to that of Vulcan. Additionally, the broad D band between 1500 and 1550 cm$^{-1}$ in Vulcan is related to the amorphous sp$^2$-bonded phase [22,23].

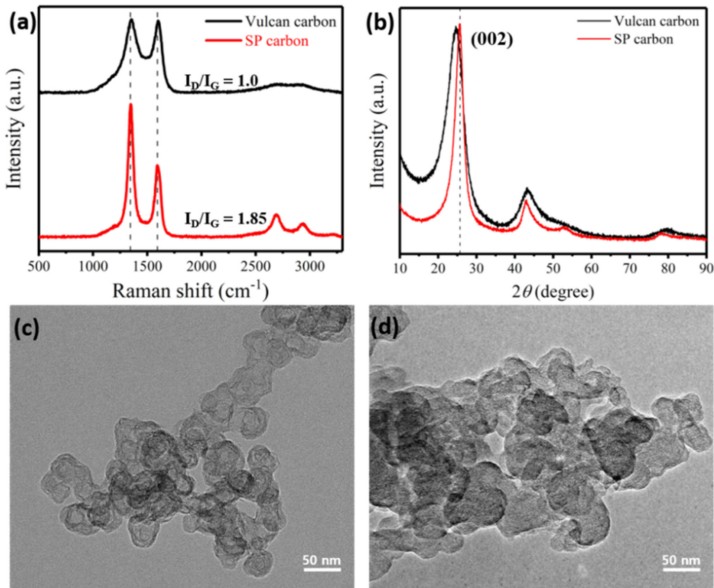

**Figure 2.** (**a**) Raman spectra and (**b**) XRD patterns of SP carbon and Vulcan XC-72. TEM images of (**c**) SP carbon and (**d**) Vulcan XC-72.

Moreover, a notable difference between the two samples is observed for the 2D band in the spectral range of 2000–3500 cm$^{-1}$ that is attributed to second-order Raman scattering. The first peak located at ~2700 cm$^{-1}$ is referred to as the 2D band, whereas the second peak at ~2950 cm$^{-1}$ is associated with the D + G combination mode and is also activated by disorder. The relative intensity ratio of the G to 2D band ($I_G/I_{2D}$) is inversely proportional to the number of graphene layers. The $I_G/I_{2D}$ ratio of SP carbon ($I_G/I_{2D}$ =3.2) is lower than that of Vulcan XC-72 ($I_G/I_{2D}$ =7.9), indicating that the latter contains fewer graphene layers. Overall, these results indicate that SP carbon contains high-grade carbonaceous materials, whereas Vulcan XC-72 contains low-grade carbonaceous materials [21,22].

Figure 2c,d shows the TEM images of SP carbon and XC-72, respectively. Both of these materials are formed by the aggregation of carbon structures with a diameter of ~20 nm, and SP carbon has a spherical morphology with a graphitic structure in Figure S2. In the X-ray powder diffraction (XRD) patterns shown in Figure 2b, both samples exhibit peaks at approximately 25.0° and 43.6° that correspond to the (002) and (101) reflections of the carbon materials, respectively. The (002) peaks for the SP carbon and Vulcan XC-72 appeared at 25.3° and 24.8°, respectively. The SP carbon is closer to the graphite peak at 26.7°, indicating that SP carbon has a graphitic structure. Thus, the XRD, Raman, and TEM results indicate that SP carbon has a graphitic structure.

### 2.3. Characterization of the Pt-TiO$_2$-SP Hybrid

To confirm the crystallinity of the Pt nanoparticles in Pt-TiO$_2$-SP, XRD measurements were performed to analyze and identify the phases in TiO$_2$-SP and Pt-TiO$_2$-SP. Furthermore, the Pt phase was compared to that of the commercial Pt/C catalyst. TiO$_2$ can adopt different structures including rutile, anatase, and brookite. Among these crystalline phases, the anatase phase has a larger band gap and lower thermal stability than the rutile phase. Since the anatase phase of TiO$_2$ is catalytically more active than the other TiO$_2$ crystalline phases, the microwave-assisted calcination was performed at 350 °C. In the XRD pattern (Figure 3), the peak positions of Pt-TiO$_2$-SP at the 2$\theta$ of 26.8°, 36.9°, 37.8°, and 38.5° correspond to the (101), (103), (004), and (112) planes of the tetragonal structure of anatase TiO$_2$, respectively [24–26] (JCPDS 21-1272). The other peaks at the 2$\theta$ of 39.5° and 46.5° correspond to the (111) and (200) planes of face-centered cubic Pt, respectively (JCPDS 04-0802). These are found in the XRD patterns of both the commercial Pt/C and the Pt-TiO$_2$-SP hybrid catalyst. The broad peak at ~25° is assigned to the (002) plane of graphitic carbon [7,8,18].

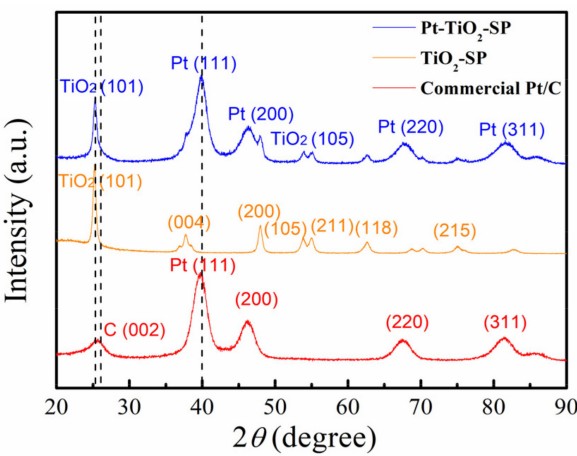

**Figure 3.** XRD patterns of Pt/C, TiO$_2$-SP, Pt-TiO$_2$-SP.

Figure 4 shows the TEM images of Pt-TiO$_2$-SP, and two different lattice fringes of nanoparticles are observed for Pt-TiO$_2$-SP in Figure 4b. One of the lattice fringes with a spacing of 0.228 nm can be attributed to Pt, whereas the other lattice fringe with a spacing of 0.332 nm can be attributed to anatase TiO$_2$ [4]. The interlayer distances of the (111) plane

of Pt and (101) plane of TiO$_2$ were calculated as 0.228 nm and 0.332 nm, respectively, from the TEM results using the Bragg equation (2dsin$\theta$ = n$\lambda$). Furthermore, the HAADF-STEM and TEM-EDS images of Pt-TiO$_2$-SP, as shown in Figure 4c–h, clearly demonstrate the distribution of the various elements, including Pt (red), C (green), O (blue), and Ti (yellow), in this hybrid catalyst. These results indicate that Pt was dispersed on the surface of TiO$_2$ and SP carbon, even though TiO$_2$ covered the carbon surface. The average size of Pt nanoparticles was determined to be 3.5 ± 1.4 nm. The EDS elemental mapping image of Pt-SP shows the distribution of Pt on carbon (Figure S3).

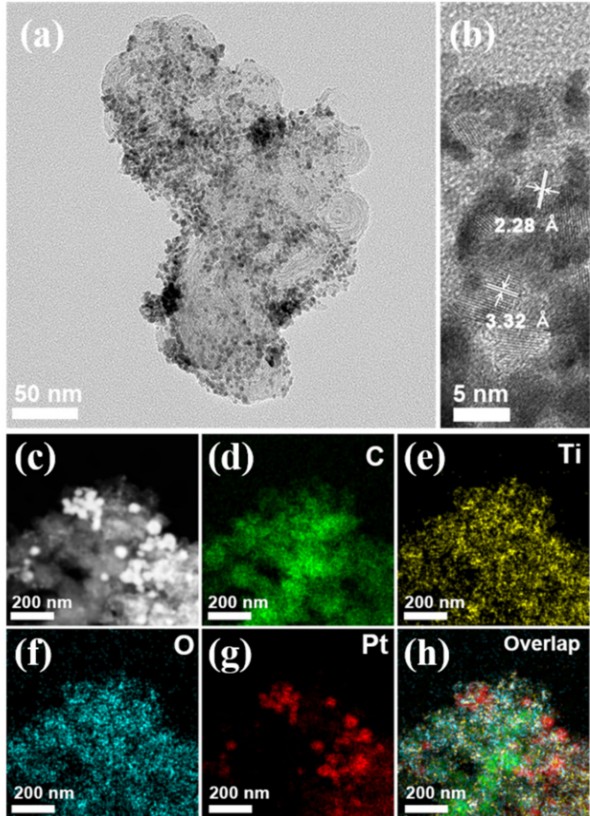

**Figure 4.** (**a**) TEM and (**b**) HR-TEM images of Pt-TiO$_2$-SP. (**c**) HAADF-STEM image and (**d–h**) the corresponding EDS elemental mapping images of Pt-TiO$_2$-SP.

X-ray photoelectron spectroscopy (XPS) was performed to investigate the chemical oxidation states of the elements and the catalytic properties of hybrid materials. The curve fittings of Pt 4f XPS for Pt-TiO$_2$-SP and Pt-SP, and Ti 2p for Pt-TiO$_2$-SP are shown in Figure 5a,b, respectively. In the Ti 2p XPS spectrum of Pt-TiO$_2$-SP, the two peaks at the binding energies of 458.2 and 463.9 eV with a splitting width of 5.7 eV are assigned to Ti 2p$_{3/2}$ and Ti 2p$_{1/2}$, respectively; they indicate the chemical state of Ti$^{4+}$ in anatase TiO$_2$ [2,27,28]. The Ti 2p peaks are slightly shifted toward higher binding energies compared to those of TiO$_2$ (at binding energies of 458.7 and 464.4 eV). Additional fitted peaks for Ti$^{3+}$ are observed in Ti 2p and in the survey spectrum (Figure S4). These are related to the partial reduction of titanium cations. These results indicate the presence of strong interactions between TiO$_2$ and carbon as well as Pt. The Pt 4f XPS spectra of Pt-TiO$_2$-SP and Pt-SP show two peaks corresponding to Pt 4f$_{7/2}$ and Pt 4f$_{5/2}$ of Pt(0). The Pt(0) peaks in Pt 4f XPS of Pt-TiO$_2$-SP are shifted to higher energies by ~0.5 eV compared to those of Pt-SP, which is attributed to an increase in the vacancy in the 5d valence band orbital of Pt. These results provide further evidence for the SMSI between Pt and TiO$_2$ in Pt-TiO$_2$-SP [3,12,14–17,29].

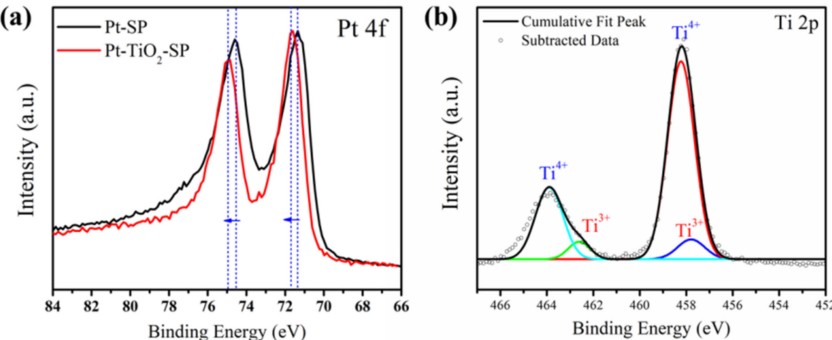

**Figure 5.** (**a**) XPS Pt 4f spectra of Pt-SP and Pt-TiO$_2$-SP catalysts, and (**b**) Ti 2p spectrum of Pt-TiO$_2$-SP.

### 2.4. Electrochemical Characterization of Pt-TiO$_2$-SP

Electrochemical measurements were performed to evaluate the catalytic performance of the Pt-SP and Pt-TiO$_2$-SP catalysts. Figure 6b shows the cyclic voltammetry (CV) curves of the Pt-TiO$_2$-SP, Pt-SP, and commercial Pt/C catalysts at 50 mV s$^{-1}$ in the scanning range of 0.05–1.0 V (vs. reversible hydrogen electrode, RHE) in N$_2$-saturated 0.1 M HClO$_4$. The ECSA values were calculated from the peak areas of the CV curves in Figure 6b using Equation (1). The ECSA indicates the electrocatalytic behavior due to the adsorption and desorption of hydrogen atoms, and was determined by integrating the CV curve between 0.05 and 0.4 V. The effective masses were obtained based on the inductively coupled plasma results that revealed the presence of 34 wt.% Pt in Pt-TiO$_2$-SP and 36 wt.% Pt in Pt-SP. The Pt-TiO$_2$-SP catalyst exhibits an ECSA of 66.1 m$^2$ g$^{-1}$, which is higher than those of Pt-SP (52.0 m$^2$ g$^{-1}$) and Pt/C (58.9 m$^2$ g$^{-1}$).

The ORR curves obtained using the rotating disk electrode (RDE) at 1600 rpm in O$_2$-saturated 0.1 M HClO$_4$ solution for Pt-TiO$_2$-SP, Pt-SP, and commercial Pt/C are presented in Figure 6a. Pt-TiO$_2$-SP and Pt-SP show more positive onset potentials than Pt/C, which is 0.9090 V, indicates that the graphitic structure enhances ORR activity. The onset potential of Pt-SP (0.9196 V) is slightly (5.9 mV) more positive than that of Pt-TiO$_2$-SP (0.9137 V), which is attributable to the low conductivity of TiO$_2$ on the SP carbon surface. The electrical conductivity was measured to investigate the effect of TiO$_2$ on the electrical conductivity of the support. The electrical conductivity of TiO$_2$-SP, SP carbon, and Vulcan are presented in Table S2. These results revealed that a TiO$_2$-SP (3.0 S cm$^{-1}$) is relatively few smaller than SP carbon (3.7 S cm$^{-1}$). It is indicating that the TiO$_2$ on carbon does not significantly affect the electronic conductivity.

Pt-TiO$_2$-SP exhibits superior catalytic ORR activity as measured by its mass activity and specific activity, as shown in Figure S5. The mass activity is 0.80 A mg$^{-1}$ for Pt-TiO$_2$-SP, 0.75 A mg$^{-1}$ for Pt-SP, and 0.69 A mg$^{-1}$ for commercial Pt/C. The values of Pt-TiO$_2$-SP is also higher than most reported Pt-based catalysts as presented in Table S1. SMSI in Pt-TiO$_2$-SP arises because the relatively electron-rich Pt nanoparticles formed on the TiO$_2$ support donate electrons to the support through Pt d-band centers; therefore, the Pt–O bond is weakened to enhance the ORR rate [14]. This result is in agreement with the results obtained previously for TiO$_2$-SP supports that are attributed to SMSI [3,4,7,15,29].

The durability of Pt catalysts was investigated by performing CV under accelerated degradation conditions in the potential range of 1.0–1.5 V at a scan rate of 500 mV s$^{-1}$. Figure 6c–f presents the changes in the ORR activities and CV curves before and after the accelerated durability tests (ADTs) for every 1000 cycles. Both catalysts exhibited a decrease in ECSA with an increase in the number of cycles. The increase in the size of the catalyst particles has been attributed to Pt migration from the surface with increasing number of cycles. One possible mechanism is the dissolution of Pt from the support into the electrolyte, and then Pt re-deposition on the surface of the support to form larger particles through a phenomenon known as Ostwald ripening. After 5000 ADT cycles, the peak area of Pt-TiO$_2$-SP catalyst in the hydrogen desorption zone decreased by 18.8% from 118.9 to

96.6 m$^2$ g$^{-1}$, whereas the ECSA of Pt-SP decreased by 66.4% from 88.4 to 58.7 m$^2$ g$^{-1}$. The electrochemical durability of the Pt-TiO$_2$-SP catalyst is slightly higher than that of the Pt-SP catalyst. Figure S6 shows the increase in the Pt particle size, confirming that the major origin of the loss of the ECSA of Pt in the Pt-SP catalyst is the corrosion of carbon and possibly Pt aggregation due to the Ostwald ripening of Pt nanoparticles. By contrast, a slight change in size was observed in the case of Pt-TiO$_2$-SP after the ADTs (Figures S6 and S7). This is because TiO$_2$ firmly anchors the Pt particles through its interactions with Pt, thereby inhibiting Pt migration and agglomeration. Meanwhile, Pt-SP exhibited a large voltage drop after 5000 cycles due to the corrosion of carbon and the subsequent detachment and agglomeration of the catalyst particles [20,30].

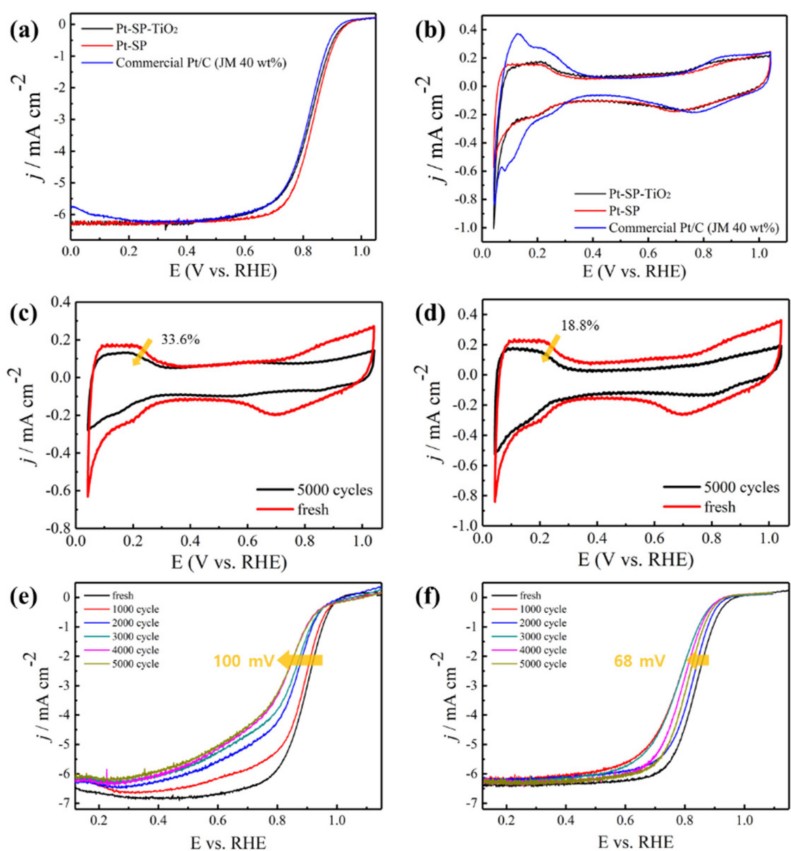

**Figure 6.** (**a**) ORR activity and (**b**) CV curves of Pt-SP, Pt-TiO$_2$-SP, and commercial Pt/C catalysts. Comparison of the electrocatalytic activity and durability, respectively, of Pt/C (**c**,**e**) and Pt-TiO$_2$-SP (**d**,**f**) before cycling (fresh) and after 5000 cycles.

## 3. Materials and Methods

### 3.1. Materials

SP carbon was obtained from Vina Tech Co., Ltd., Jeonju-si, Korea. Vulcan XC-72 carbon (Cabot corporation), titanium (IV) tetrachloride (TiCl$_4$, 99%, Wako Chemicals, Japan), ethylene glycol (99.8%, Sigma-Aldrich, headquarters, Burlington, MA, USA), isopropyl alcohol (IPA, 99.5%, Sigma-Aldrich), chloroplatinic acid hexahydrate (H$_2$PtCl$_6$ 6H$_2$O, Sigma-Aldrich), sodium hydroxide (NaOH, 97%, Sigma-Aldrich), and perchloric acid (HClO$_4$, 70%, Sigma-Aldrich) were purchased. All of the chemicals were used as received without further purification.

### 3.2. Synthesis of TiO$_2$-SP Carbon Composite

The composite of SP carbon and TiO$_2$ was prepared by a microwave-assisted hydrothermal reaction. SP carbon (0.1 g) was dispersed in deionized water (40 mL) in a Teflon

beaker. Then, $TiCl_4$ (2 M solution in deionized water; 0.4 mL) was added under vigorous stirring. The mixing temperature was increased up to 80 °C and the sample was heated in a microwave oven at a power of 200 W for 20 min. After the microwave process, the resultant mixture was filtered, washed with deionized water and ethanol, and dried overnight in an oven at 80 °C. The resultant mixture was heat-treated at 350 °C for 1 h to form anatase $TiO_2$. Thereafter, $TiO_2$-SP (60 mg) was dispersed in ethylene glycol (100 mL) and sonicated for 10 min. Subsequently, $H_2PtCl_6$ (84 mg) was added to the mixture under vigorous stirring for 10 min and the pH of the suspension was adjusted to 12 using NaOH; then, the mixture was stirred continuously for 60 min. The suspension was then heated for 3 h at 160 °C. Thereafter, the suspension was filtered and washed with ethanol and distilled water. The obtained Pt-$TiO_2$-SP was finally dried overnight at 80 °C. To compare catalyst performance, Pt-SP was synthesized by the same method except for the inclusion of $TiO_2$.

### 3.3. Characterization

XRD patterns of the crystalline solids were recorded in the $2\theta$ range of 10–90° at room temperature on an X-ray diffractometer (D/max 2200 V/PC, Rigaku, Japan) equipped with a Cu *Kα* radiation source ($\lambda$ = 1.5406 Å) and diffracted beam monochromator at 40 kV and 40 mA. Low- and HR TEM images were obtained using a Philips Tecnai G2 F30 Super-Twin transmission electron microscope operated at 300 kV. XPS was performed using a Thermo VG Scientific Sigma Probe spectrometer using a Al *Kα* X-ray ($\lambda$ = 1486.6 eV) as the excitation source. The XPS data were calibrated using the C 1s peak at 284.5 eV. Raman spectra were obtained using a Jobin Yvon/Horiba Lab RAM spectrometer with an Olympus BX 41 integral microscope. The 632.8 nm light from an air-cooled He/Ne laser was used as the excitation source. The laser beam was focused onto a spot with a diameter of ~2 μm on the sample with an objective microscope and the spectra were recorded in the range of 100–2000 $cm^{-1}$. The electrical conductivity of the powder samples was measured at 150 MPa by Teflon cylinder using a current-voltage test with a VSP potentiostat (biologic).

### 3.4. Electrochemical Performance

CV and ORR activity measurements were performed in a three-electrode cell system using a ZIVE BP2A model potentiostat and Wave Vortex rotating ring disk electrode model. The electrocatalytic activity of the hybrid samples was evaluated using a glassy carbon electrode (5 mm diameter), Ag/AgCl (3 M NaCl), and a Pt wire as the working, reference, and counter electrodes, respectively. The catalyst ink was prepared by mixing the catalyst (3.0 mg), distilled water (280 μL), isopropyl alcohol (200 μL), and a 5 wt% Nafion solution (20 μL) under sonication for 1 h. Then, the catalyst (3 μL) was deposited on the working electrode, and the working electrode was dried.

To remove the remaining Nafion from the catalyst surface and activate the RDE, the working electrode was first electrochemically cleaned by subjecting it to 100 cycles in the potential range of 0.05–0.8 V vs. RHE at a scan rate of 50 mV $s^{-1}$. CV analysis was performed by sweeping the potential between 0.05 and 1.10 V vs. RHE at a scan rate of 20 mV $s^{-1}$. Furthermore, to study the durability of the catalysts, an accelerated durability test was conducted by cycling the working electrode between 0.6 and 1.0 V vs. RHE at 500 mV $s^{-1}$ in a $N_2$-saturated electrolyte for 5000 cycles to simulate the peak power to idle transients in an automotive drive cycle. The ORR activity measurement and CV were taken every 1000 cycles. The ECSA of Pt can be calculated from the Coulombic charges accumulated during hydrogen adsorption or desorption after correcting for the double-layer charging current from the CVs as follows:

$$ECSA = \frac{Q_H}{0.21 \times M_{Pt}} \tag{1}$$

where $Q_H$ (mC) is the charge due to the hydrogen adsorption/desorption in the hydrogen region of the CV, and 0.21 mC $cm^{-2}$ is the electrical charge associated with the monolayer adsorption of hydrogen on Pt. $M_{Pt}$ is the mass of Pt loaded on the working electrode [4].

The mass specific activity ($I_m$) was calculated using the limiting current ($I_{lim}$) at 0.2 V vs. RHE and current ($I$) at 0.9 V as follows:

$$I_m = \frac{I_{lim}I}{(I_{lim} - I)M_{Pt}} \tag{2}$$

The specific activities ($I_s$) were measured by normalizing current to the *Pt ECSA* as follows:

$$I_s = \frac{I_{lim}I}{(I_{lim} - I)ECSA} \tag{3}$$

## 4. Conclusions

We synthesized a $TiO_2$-SP carbon hybrid support for Pt via a simple microwave-assisted method. The Pt nanoparticles on $TiO_2$ showed improved electrocatalytic activity. SP carbon with a high degree of graphitization was investigated as electrocatalyst support. The synthesized Pt-SP-$TiO_2$ electrocatalyst has a uniform distribution of $TiO_2$ and Pt particles with the sizes of 5 nm and 3–5 nm, respectively. Electrocatalytic ORR activity and durability tests showed that Pt-$TiO_2$-SP exhibits better corrosion resistance and durability than Pt-SP because of the SMSI effect. The results obtained for this hybrid catalyst can be attributed to two key factors: (1) the use of a graphitic carbon support and (2) SMSI between the Pt nanoparticles and $TiO_2$. Moreover, after ADTs the loss in ECSA for Pt-$TiO_2$-SP was significantly less than that for Pt/C, with a decrease of only 18.8% from the original ECSA obtained for Pt-$TiO_2$-SP, due to the SMSI. These results indicate that the enhanced electrocatalytic activity of Pt particles supported on the hybrid support can potentially lead to their high catalytic activity in PEMFCs [8,31,32].

**Supplementary Materials:** The following are available online at https://www.mdpi.com/article/10.3390/catal11101196/s1, Figure S1: Nitrogen adsorption-desorption isotherms for Vulcan carbon and SP carbon; Figure S2. HR-TEM images of SP carbon; Figure S3. (a) TEM, (b) magnification of TEM and (c) HR-TEM images of SP-Pt. (d) HAADF-STEM image and (e-g) Corresponding elemental mapping images; Figure S4. XPS survey spectra of Pt-SP, Pt-$TiO_2$, and Pt-$TiO_2$-SP; Figure S5. Mass activity and specific activity of SP-$TiO_2$, SP-Pt, and commercial Pt/C; Figure S6. TEM images of before (a,b) and after 5000 cycles (c,d) of Pt-$TiO_2$-SP (a,c) and commercial Pt/C (b,d); Figure S7. (a) TEM images of after 5000 cycles 40 wt.% Pt on SP-$TiO_2$. Corresponding elemental mapping images (b–g); Table S1: ORR performances for the present Pt-$TiO_2$-SP and the reported Pt-based catalyst at 0.9 V; Table S2. Electrical conductivity of SP carbon, $TiO_2$-SP, Vulcan carbon, and $TiO_2$.

**Author Contributions:** Conceptualization, S.-J.J. and K.C.R.; formal analysis J.-S.H. and Y.W.L.; data curation T.H.S. and Y.C.K.; writing—original draft preparation S.-J.J.; writing—review and editing T.H.S., Y.W.L., Y.C.K. and K.C.R. All authors have read and agreed to the published version of the manuscript

**Funding:** This research received no external funding.

**Acknowledgments:** This research was supported by the Basic Science Research Program of the National research Foundation of Korea (NRF-202006610001). This work was supported by the Korea Institute of Energy Technology Evaluation and Planning (KEPET) grant funded by the Korea government (MOTIE) (20183030032010). This work was supported by Korea Institute of Energy Technology Evaluation and Planning (KETEP) grant funded by the Korea government (MOTIE) (20203030030070). This research was supported by the National Research Foundation of Korea (NRF) grant funded by the Korea government (MSIT) (2020M3H4A3105824).

**Conflicts of Interest:** The authors declare no conflict of interest.

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
