# Peer review of "Hybrid Structure of TiO2-Graphitic Carbon as a Support of Pt Nanoparticles for Catalyzing Oxygen Reduction Reaction"

_catalysts, doi:10.3390/catal11101196_

Round 1

Reviewer 1 Report

This manuscript reports a synthesis of a hybrid support consisting of SP carbon and TiO2 for Pt nanoparticles. The authors synthesized Pt-TiO2-SP carbon hybrid catalyst through a simple micro-wave-assisted method and characterized them. The prepared catalyst indicated improved electrocatalytic activity and durability through ORR, which can be attributed to graphitic carbon support and SMSI effect. The study is relatively complete but there are some questions to be solved before publications.

  1. On page 1, I assume “manifests” and “indicates” are repeated and on page 9, what does “were c again” mean?
  2. In figure 2. TEM images(c,d), I can’t clearly see the spherical morphology of synthesized SP carbon compared with Vulcan XC-72, there aren’t HR-TEM images to characterize their structures.
  3. In figure 4. EDS elemental mapping images(c-h) do not show that Pt was homogeneously dispersed on the surface of TiO2 and SP carbon.
  4. In electrochemical characterization, why figure 6. (a) shows that Pt-SP catalyst had better specific activity than Pt-TiO2-SP catalyst? It contradicts with the highest activity of Pt-TiO2-SP catalyst.
  5. The authors declare that Pt-TiO2-SP exhibits superior catalytic mass activity, but there is no comparison with mass activity of commercial Pt/C. Half-wave potentials can also be used to show the performance of catalysts.
  6. There are no tafel slopes or K-L plot to characterize the kinetics of ORR.
  7. In figure 6. (e,f), the results of accelerated degradation tests do not sufficiently provide evidence on the durability. Authors should show corresponding ECSA data of catalysts after 5000 cycles.
  8. The results of ADTs do not show a good durability, and there are no comparisons of the data with commercial Pt/C catalyst, nor with the benchmarks.
  9. In figure S3 and 4, why agglomeration of Pt-SP catalyst after ADTs is severe than Pt-TiO2-SP catalyst but Pt-TiO2-SP catalyst shows better durability?
  10. Why the accelerated degradation tests are measured in N2-saturated electrolyte rather than in O2-saterated electrolyte, as reported in most papers.
  11. The authors declare that “a slight change in size was observed in the case of Pt-TiO2-SP after the ADTs (Figure S3)”, but in SI, it is corresponded to figure S4.
  12. The authors are suggested to check and correct some typo and figure errors.

Reviewer 2 Report

The manuscript “Hybrid structure of TiO2-graphitic carbon as a support of Pt nanoparticles for catalysing oxygen reduction reaction” by Su-Jin Jang, Yun Chan Kang, Jin-Su Hyun, Tae Ho Shin, Young Wook Lee, and Kwang Chul Roh describes research focused on a Pt catalyst supported on a TiO2-graphitic carbon substrate and the use of this catalyst in the ORR reaction. Standard methods such as TEM, SEM, Raman, XPS, and XRD were used for the material characterization of the catalyst. The electrochemical activity and durability of the catalyst were tested using CV and LSV. This is a well written article and the results are clearly shown. However, catalysts with more than 30% wt. Pt content are not considered attractive at present. Currently, there is a tendency to design catalysts based on the idea of single atoms and to replace Pt with other materials with lower cost and greater stability. Moreover, a table comparing the obtained results with the literature (onset potential, loading, stability, etc) should be provided. I also found a lot of typing errors. Considering the above, I believe that this manuscript is not suitable for publication in Catalyst.

Round 2

Reviewer 1 Report

The revisions are fine.

Reviewer 2 Report

The authors responded to only one of my critical remarks and added tables comparing their results with the literature. Additionally, I noticed that the manuscript lacked information about the Pt content in a commercial Pt/C sample.  I still believe that the manuscript does not present any new solutions and probably if the amount of Pt was increased, the obtained catalyst would show even better catalytic properties. 

Author Response

Answer: First, we greatly appreciate and agree with the reviewers’ thoughtful comment. When manufacturing PEMFC, fuel cell performance is good in case the Pt content is more than 30 wt.%. We develop this catalyst and its performance tested with the aim of developing MEA fabrication.

(RSC Adv., 2021, 11, 19417–19425, Xie, L., Kirk, D.W. Electrocatalysis 11, 292–300 (2020), Journal of Power Sources 489 (2021) 229485, Adv. Mater. 2021, 33, 2007885, J. Mater. Chem. A, 2019, 7, 25056–25065)